# Highlights on the Luspatercept Treatment in Thalassemia

Yesim Aydinok

Department of Pediatric Hematology, Ege University Medical School, 35100 Izmir, Turkey;
yesim.aydinok@yahoo.com

**Abstract:** Luspatercept has been shown to act as a ligand trap, selectively suppressing the deleterious effects of GDF11 that blocks terminal erythroid maturation, restoring normal erythroid differentiation and improving anemia in animal models of β-thalassemia. Effective doses of luspatercept achieved hemoglobin increase within 7 days of the first dose, and plasma half-life supports subcutaneously administration every 21 days in adults with β-thalassemia. A Phase 3, placebo-controlled 1-year study with starting dose of 1.0 up to 1.25 mg/kg every 21 days achieved ≥33% reduction in red cell transfusion volume in 21.4% of adult transfusion-dependent β-, HbE/β-thalassemia patients on luspatercept vs. 4.5% on placebo over a fixed 12-week period, and 41.1% of patients in luspatercept vs. 2.7% placebo in any 24-week period. Luspatercept allowed ≥1.0 and ≥1.5 g/dL increase in hemoglobin from baseline in 77% and 52.1% of adult non-transfusion-dependent β-, HbE/β-thalassemia patients vs. 0% placebo over a 12-week interval. Although not significant, a greater improvement in patient-reported outcomes was observed with luspatercept. Luspatercept had a manageable safety profile with notable adverse effects of venous thromboembolism in 3.6% of transfusion-dependent β-thalassemia vs. 0.9% of placebo and extramedullary hematopoiesis in 6% of non-transfusion-dependent β-thalassemia vs. 2% of placebo. The pediatric study started patients' enrollment.

**Keywords:** luspatercept; transfusion-dependent thalassemia; non-transfusion-dependent thalassemia; ineffective erythropoiesis; beta thalassemia; HbE/beta thalassemia





## 1. Introduction

Regular transfusion programs combined with appropriate iron chelation therapies prolonged and normalized life in transfusion-dependent thalassemia (TDT). Hematopoietic stem cell transplantation has been a curative therapy for pediatric TDT patients for many years. However, it has limitations of HLA identical donor availability and age. Gene addition and gene editing therapies have achieved transfusion independence, but it is expensive and excludes patients with high iron burden and pre-existing organ dysfunction. Therefore, transfusion dependence and its consequence, iron overload, and iron-induced morbidity and mortality have remained the major problems in disease management in TDT [1]. Lower hemoglobin (Hb) levels define patients at risk of developing morbidity in NTDT [2]. There is an unmet need for novel treatments, disease-specific HRQoL, and favorable disease management approaches to decrease the burden of TDT. Non-transfusion-dependent thalassemia (NTDT) was found to have far worse HRQoL compared with TDT [3].

A better understanding of the pathogenesis of thalassemia has stimulated preclinical research on alternative or concomitant novel therapies that include correcting the globin chain imbalance, addressing ineffective erythropoiesis, decreasing hemolysis, or improving iron dysregulation, thereby ameliorating ineffective erythropoiesis [4]. Some of these approaches have progressed to clinical research. Agents such as phosphodiesterase 9 (PDE9) inhibitors increasing cyclic guanosine monophosphate (cGMP) levels may stimulate the production of HbF and improve globin chain imbalance. However, a Phase 2b Clinical Trial with a PDE9 inhibitor tovinontrine (IMR-687) in both TDT and NTDT cohorts (NCT04411082) demonstrated no meaningful benefit in transfusion burden or improvement in most disease-related biomarkers and terminated early [5]. Inhibiting EPO/Jak2

pathway by ruxolitinib might decrease the erythroid expansion and reverse splenomegaly. In turn, this might reduce the number of sequestrated red cells in the spleen and the blood transfusion requirement. A Phase 2a study of ruxolitinib in TDT demonstrated a steady decrease in spleen length but limited effect on transfusion needs. Further development of ruxolitinib was not planned for thalassemia [6]. Ineffective erythropoiesis (IE) is a hallmark of the pathophysiology of transfusion-dependent (TDT) and non-transfusion-dependent thalassemia (NTDT). Downregulation of hepcidin and, consequently, upregulation of ferroportin in thalassemia has a subsequent impact on iron homeostasis resulting in increased iron absorption and erythroid iron intake, ensuing oxidative damage [7]. Another group of agents targeting iron dysregulation can break the vicious cycle promoting IE, and improve anemia. Two consecutive Phase 2 studies have been conducted with weekly subcutaneous injections of synthetic human hepcidin in TDT (www.clinicaltrials.gov at (accessed on 29 July 2021) #NCT03381833) and minihepcidin by dose escalation in TDT and NTDT for treating chronic anemia (NCT03802201). However, both studies showed no clinically meaningful activity [8]. A Phase 2a clinical study with ferroportin inhibitor vamifeport (VIT-2763) in NTDT pa-tients reduced iron levels in circulation after drug administration demonstrating its target engagement with ferroportin [9]. Transmembrane protease serine 6 (TMPRSS6) is predominantly expressed in liver cells and suppresses hepcidin expression. Therefore, TMPRSS6 inhibition attenuates the negative regulation of hepcidin, resulting in increased hepcidin expression. A Phase 2a Clinical Trial with ISIS 702843 (TMPRSS6-LRx), an antisense oligonucleotide (ASO) drug targeted to TMPRSS6, in NTD-β-Thalassemia is currently being conducted (www.clinicaltrials.gov at (accessed on 26 September 2022) # (NCT04059406). Another agent, small interfering RNAs specifically targeting TMPRSS6 mRNA, has been shown to improve anemia and iron overload in preclinical models and is currently under clinical development (www.clinicaltrials.gov at (accessed on 1 April 2022) (NCT04718844) [10]. Mitapivat, a pyruvate kinase activator, represents a unique mechanism to mitigate ineffective erythropoiesis and improve anemia. Preliminary results of a Phase 2 trial of mitapivat in α- and β- NTDT showed a promising response in hemoglobin [11] and proceeded to the Phase 3 study of mitapivat in TDT and NTDT (www.clinicaltrials.gov at (accessed on 13 February 2023) #NCT04770779, # NCT04770753). are currently conducting.

Although the medications modulating disease severity in TDT and NTDT are grounded in strong preclinical research, most of them have not been clinically impactful in improving anemia and reducing the transfusion burden. In this review, we focus on the modified activin receptor IIB ligand trap, luspatercept (ACE-536), as an erythrocyte maturating agent for the treatment of anemia in patients with TDT and NTDT that has achieved the study endpoints by decreasing transfusion volume in a substantial portion of the patients with TDT, and provided a meaningful increase in hemoglobin in NTDT.

## 2. Ineffective Erythropoiesis in β-Thalassemia

In β-thalassemia, the ineffective production of red cells (RBCs) has been attributed to the accumulation of alpha-globin chains, leads to the formation of aggregates, which generates proteotoxicity and impairs erythroid maturation, triggering apoptosis [12,13]. Experimental data suggested that different mechanisms, including decreased differentiation combined with increased proliferation and apoptosis of erythroid progenitors, are involved limited production of RBCs in thalassemia [14,15]. In thalassemia, hypoxia induces the production of erythropoietin (EPO), which interacts with EPO receptor (EPOR), resulting in persistent phosphorylation of Janus kinase 2 (Jak2)/STAT5 pathway, hence, proliferation and expansion of early erythroid progenitors [16,17]. It has demonstrated that expanded erythroid progenitors in thalassemia overexpress GDF11 that autoinhibit terminal erythroid maturation by binding Activin receptor II on erythroid cells and activation of phosphorylated SMAD2/3 resulting in apoptosis at the polychromatophilic stage and a suboptimal production of RBCs [18,19] leading to vicious cycle (Figure 1).

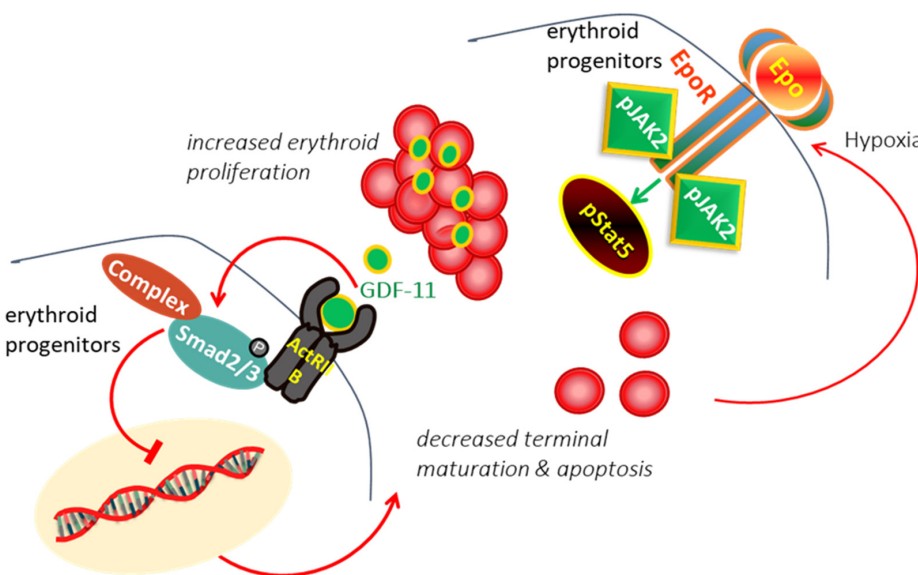

**Figure 1.** In β-Thalassemia, the hypoxia arising from the lack of production of normal RBC induces excessive EPO production. EPO act through EPO/EPOR/JAK2/STAT5 pathway to force the proliferation of erythroid progenitors. The expanded erythroid progenitors overexpress GDF11 that autoinhibits terminal erythroid maturation by binding Activin receptor II on erythroid cells resulting in apoptosis and ineffective erythropoiesis.

## 3. Modulating Erythroid Differentiation by ActRII Ligand Traps

The TGF-β superfamily ligands, including GDF11, GDF8, BMP6, BMP10, and Activin B, bind to specific type I and type II transmembrane receptors on erythroid lineage, leading to phosphorylation of SMAD proteins that play a critical role in regulating normal and stress erythropoiesis [20].

Two different recombinant receptor fusion proteins that consist of the extracellular domain of either ActRIIA (sotatercept, ACE-011) or ActRIIB (luspatercect, ACE-536) linked to the Fc protein of human immunoglobulin (IgG) have been developed and shown to act as ligand traps that selectively captures and neutralizes GDF11, and inhibit signaling through SMAD2/3, thereby, restoring normal erythroid differentiation [18,19]. Experimental studies using the murine analog of luspatercept (RAP-536) demonstrated that luspatercept-mediated inhibition of SMAD2/3 enhanced late-stage erythropoiesis by restoring nuclear levels of the transcription factor GATA-1 and up-regulation of HSP70 in erythroid precursors leading to amelioration of oxidative stress and promotion of erythroid differentiation [21]. Terminal maturation of red cells by RAP-536 mitigated IE and reduced hemolysis. It alleviated anemia and reduced comorbidities associated with IE in β-thalassemia, such as decreased bone mineral density, splenomegaly, and iron overload in the mouse model [22].

## 4. Pharmacokinetics of Luspatercept

Subcutaneous (s.c) administration of luspatercept is linearly absorbed into the circulation. It reaches the Cmax in the serum approximately seven post-dose days across all dose levels in healthy volunteers and patients. The large molecular mass of luspatercept (MW; 75.7 kDa) allows its distribution in extracellular fluids. Population studies suggest that Cmax and AUC in serum increase proportionally at doses from 0.125 to 1.25 mg/kg, while clearance is independent of the dose. Luspatercept is assumed to be catabolized into amino acids by a general protein degradation process. β-thalassemia patients showed a 36% inter-individual variability in AUC. The mean half-life in serum was 11 days for β-thalassemia patients. Hb response to luspatercept occurred within 7 days of the first dose, and the increase correlated with the time to reach Cmax. The greatest mean Hb increase was observed after the first dose, with additional smaller increases observed after subsequent doses when administered every three weeks. Hb levels returned to baseline value

approximately 6 to 8 weeks from the last dose. The exposure-Hb response relationship provided the rationale for a starting dose of 1 mg/kg that was enough for most of the early responders (71–80%), and a further dose increase to 1.25 mg/kg increased the number of responders by at least 20%. Overall, the starting dose of 1 mg/kg and dose escalation to 1.25 mg/kg was sufficient for achieving meaningful and sustained reductions in RBC transfusion burden. Population pharmacokinetics demonstrated no clinically significant difference in mean steady-state Cmax, AUC, and clearance of luspatercept by race, age, and hepatic and renal impairment [23–25].

## 5. Clinical Studies with Luspatercept

A Phase 1 clinical study of ACE-536 (www.clinicaltrials.gov at (accessed on 7 November 2012) #NCT01432717) was designed as a randomized, double-blind, placebo-controlled, dose-escalation trial in which 32 healthy volunteers received two s.c. doses of either luspatercept (0.0625–0.25 mg/kg) or placebo every 2 weeks. The treatment was generally safe and well tolerated. ACE-536 increased Hb levels in a dose-dependent manner. The PK profile of ACE-536 demonstrated a long-acting pattern of s.c. exposure supporting the use every 3 weeks [24].

A Phase 2, open-label, nonrandomized, uncontrolled ascending dose study of ACE-536 (www.clinicaltrials.gov (accessed on 14 December 2016) at #NCT01749540) was conducted in 64 adult patients with 31 TDT (≥4 RBC Units transfused per 8 weeks) and 33 NTDT (Hb < 10 g/dL, <4 RBC Units per 8 weeks). Patients received 0.2 to 1.25 mg/kg luspatercept s.c. every 3 weeks for five cycles (dose-finding stage) and 0.8 to 1.25 mg/kg for a 5-year extension period (www.clinicaltrials.gov (accessed on 19 May 2021) at #NCT02268409). Ascending dose study demonstrated that luspatercept doses of <0.6 mg/kg were not efficacious. Effective doses of luspatercept achieved improvement in Hb or transfusion burden reduction with an acceptable safety profile supporting a randomized clinical trial to assess efficacy and safety [26]. A Phase 2, open-label, dose-finding study of ACE-011 (www.clinicaltrials.gov (accessed on 21 November 2022) at #NCT01571635) was also conducted in adult β-thalassemia patients with positive findings. Still, the sponsor decided not further development of sotatercept in β-thalassemia [27].

A Phase 3, randomized, double-blind, placebo-controlled, multicenter study (www.clinicaltrials.gov (accessed on 25 January 2022) at # NCT02604433) was designed as a 1-year study of luspatercept/placebo (2:1) with a 5 years open-label follow-up in adult TDT patients [BELIEVE trial]. Starting dose was 1 mg/kg up to 1.25 mg/kg subcutaneously every 21 days. The 1-year placebo-controlled study (luspatercept *n* = 224: placebo *n* = 112) met primary endpoints demonstrating a statistically significant reduction in transfusion burden by at least 33% in 21.4% of patients on luspatercept vs. only 4.5% of patients on placebo over a fixed 12-week period. Furthermore, 71 and 41% of patients at any 12- and 24-week periods, respectively, demonstrated a statistically significant and clinically meaningful reduction (≥33% response) in RBC transfusion with luspatercept vs. placebo [28] that has been shown to persist with longer-term treatment in the 5-year open-label extension phase of BELIEVE trial patients [29]. Further, the durable reductions in RBC transfusions resulted in decreased iron chelation use and improvement or stabilization of iron-related parameters by luspatercept over time [30]. The median time to the first response among patients with ≥33% reductions in the transfusion burden during any 12-week interval was within the first luspatercept treatment cycle, and 75% of patients had a response within four treatment cycles. The clinical benefit of luspatercept was observed in all patient subgroups stratified per age, gender, region, and baseline transfusion burden. Subgroup analysis suggested that the percentage of patients who responded to luspatercept by ≥33% reductions in the transfusion burden may be lower in patients with a β0/β0 genotype than in those with a non-β0/β0 genotype. Furthermore, post hoc analysis indicated that the response was faster among the patients with a non-β0/β0 genotype than among those with a β0/β0 genotype [28]. However, the long-term efficacy analysis of luspatercept in the 5-year open-label extension phase of BELIEVE trial patients with β0/β0 genotypes experienced similar reductions in

RBC transfusion burden with luspatercept as the overall study population suggesting the efficacy of luspatercept in patients with more severe disease [31]. Luspatercept was generally well tolerated, with primary adverse events (AEs) including arthralgia, bone pain, dizziness, hypertension, and hyperuricemia. Venous thromboembolism (VTE) was also noted in 3.6% of patients on the luspatercept, including deep vein thrombosis, cerebral venous sinus thrombosis, portal vein thrombosis, post-thrombotic syndrome, pulmonary embolism, and thrombotic stroke vs. 0.9% (phlebitis) placebo arm. All such events occurred in patients who had undergone splenectomy and had at least one risk factor for thromboembolic disease at baseline [28]. The long-term safety analysis of the BELIEVE trial patients treated with luspatercept (*n* = 315) during the median of 103 weeks (range 1.7–215 weeks) was consistent with the safety profile of luspatercept with no new safety concerns. VTE occurred in 4.1% vs. 0.9% of patients in the luspatercept vs. placebo arms, respectively [32].

A Phase 2, double-blind, randomized, placebo-controlled, multicenter study (www. clinicaltrials.gov (accessed on 19 December 2022) at # NCT03342404) was designed as a 1-year study of luspatercept/placebo (2:1) in 145 adult patients with NTDT and a Hb level $\leq$ 10 g/dL. The study met the primary endpoint with 77% of patients who had a mean increase in Hb $\geq$ 1.0 g/dL from baseline in the luspatercept vs. 0% placebo arm, and the secondary endpoint with 52.1% of patients achieved a Hb increase of $\geq$1.5 g/dL from baseline in the luspatercept vs. 0% placebo arm in over a continuous 12-week interval (weeks 13–24), in the absence of RBC transfusion. Patients treated with luspatercept had higher improvement in the NTDT-PRO tool Tiredness/Weakness domain score, compared with placebo, which, although not significant, was more pronounced over time. Post hoc analyses have shown that a higher increase in Hb from baseline was associated with a greater improvement in the NTDT-PRO tool Tiredness/Weakness domain score. The most common treatment-emergent AEs were bone pain, headache, and arthralgia. Notably, no thromboembolic events were reported in patients treated with luspatercept or placebo during a relatively short follow-up period of the median of 99.7 weeks (range 77.0–117.5) in the luspatercept and 61.1 weeks (37.1–96.9) in the placebo group. Extramedullary hematopoiesis masses were observed in 6% of patients treated with luspatercept vs. 2.0% of patients treated with placebo [33].

Phase 2, an open-label study evaluating the safety and pharmacokinetics of luspatercept in pediatric patients with TDT (www.clinicaltrials.gov (accessed on 13 February 2023) at # NCT 04143724), designed to determine recommended doses for the 12– < 18 years, was started patients' recruitment. The patients with 6– < 12 years will be enrolled after completing a 1-year study in the 12– < 18 age cohort and reviewing comprehensive safety data. Both groups will be followed on treatment for up to 5 years [34] (Table 1).

**Table 1.** Summary of completed or ongoing studies of the luspatercept clinical program.

| Protocol no | Phase | Planned/Treated | Dose (mg/kg) | Duration | Erythroid Response |
|---|---|---|---|---|---|
| NCT01432717 | 1 | 32 PM women | 0.0625–0.25 | 1 month (Q2W) | Hb $\geq$ 1 g/dL, 83% (0.25 mg/kg) |
| NCT01749540 | 2 | 64/64 (NTDT, TDT, $\geq$18y) | 0.2–1.25 | 15 weeks (Q3W) | $\int$ NTDT; 58% achieved Hb $\geq$ 1.5 g/dL for 2 weeks, 71% Hb $\geq$ 1.0 g/dL for 12 weeks. $\int$ TDT; transfusion reduction $\geq$ 20, 33, and 50% at 81, 72, and 63%, respectively, for 12 weeks |
| NCT02268409 | 2 | 64/51 (NTDT, TDT, $\geq$18y) | 0.8–1.25 | 60 months (Q3W) | |
| NCT02604433 [BELIEVE] | 3 | 336/332 (TDT, $\geq$18y, luspatercept vs. placebo 224:112) *open-label LTFU ongoing* | 1.0–1.25 | 48 weeks LTFU:60 months (Q3W) | $\geq$33% reduction: weeks 13–24; 21.4% in luspatercept vs. 4.5% placebo $\geq$50% reduction: weeks 13–24; 7.6% in luspatercept vs. 1.8% placebo and $\geq$50% reduction: weeks 37–48; 10.3% in luspatercept vs. 0.9% placebo $\geq$33% reduction: any 24 weeks; 41.1% in luspatercept vs. 2.7% placebo |

**Table 1.** *Cont.*

| Protocol no | Phase | Planned/Treated | Dose (mg/kg) | Duration | Erythroid Response |
|---|---|---|---|---|---|
| NCT03342404 [BEYOND] | 2 | 150/145 (NTDT, ≥18y, luspatercept vs. placebo 96:49) | 1.0–1.25 | 48 weeks (Q3W) | Hb ≥ 1.0 g/dL: weeks 13–24; 77% in luspatercept vs. 0% placebo ($p < 0.0001$) Mean Hb change (g/dL) from baseline weeks 13–24; 1.48 in luspatercept vs. 0.07 placebo ($p < 0.0001$) Hb ≥ 1.0 g/dL: weeks 37–48; 71% in luspatercept vs. 2% placebo ($p < 0.0001$) |
| NCT04143724 | 2 | 54/- (TDT, 6– < 18y) | 0.75–1.25 | 1 year LTFU: 5 year | Started patients' recruitment |

∫ only patients receiving dose levels of 0.6–1.25 mg/kg of luspatercept were accounted for. PM; Post-menopausal, Hb; hemoglobin, TDT; transfusion-dependent thalassemia, NTDT; non-transfusion-dependent thalassemia, LTFU; long-term follow-up.

## 6. Current Status, Challenges, and Future Directions of Luspatercept

The U.S. Food & Drug Administration (FDA) and the European Medicines Agency (EMA) approved luspatercept for the treatment of anemia in adult patients with transfusion-dependent β-thalassemia in November 2019 and June 2020, respectively, by considering the risk: benefit ratio, as luspatercept reduces the need for blood transfusions with manageable side effects. The efficacy and safety profile in pediatric patients are required to be established.

Standard of care including regular transfusion and chelation in TDT can cause low satisfaction, low compliance, with possible negative consequences of health-related quality of life (HrQoL), and costs. The long-term real-life data will reveal the impact of luspatercept use on HRQoL, at least, in a group of TDT patients. Although a noticeable improvement in patient-reported outcomes was noticed in NTDT patients receiving luspatercept than in patients receiving placebo, it was not significant [33]. Longer-term follow-up with luspatercept will further explore whether improvement in hemoglobin concentrations may ameliorate clinical complications and HRQoL in NTDT. The results from luspatercept studies exploring its mechanism of action implicate promoting terminal erythroid differentiation and mitigating IE, thereby ameliorating anemia [18,22]. Therefore, a higher rate of extramedullary hematopoietic masses in patients receiving luspatercept, an agent directly targeting IE, remains to be elucidated. Further research is needed to define the exact mechanism of action of luspatercept and clarify the consequences of TGF-beta inhibition at the cellular level.

## 7. Conclusions

Several agents targeting α/β globin chain imbalance, ineffective erythropoiesis, or iron dysregulation are currently being investigated in clinical trials for modifying disease severity in thalassemia. Luspatercept has been the first novel approved therapy that may reduce transfusion requirements and represents an exciting new therapeutic option beyond the standard of care in adult patients with transfusion-dependent β-thalassemia.

**Funding:** This research received no external funding.

**Conflicts of Interest:** Member of the advisory board BMS, provided consultancy to CRISPR-Vertex Therapeutics and Silence Therapeutics, received research funding from BMS, Novartis, Ionis Pharmaceuticals, Resonance Health, and Imara.

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
