# Peer review of "Highlights on the Luspatercept Treatment in Thalassemia"

_thalassrep, doi:10.3390/thalassrep13010008_

Round 1

Reviewer 1 Report

Is a well-written review regarding the use luspatercept in thalassemia. I think that it is suitable for publication. 

Minor points:

I think that a minor comment about the mechanism of action of luspatercept should be added.  It causes up-regulation of HSP70 and leads to amelioration of oxidative stress and promotion of erythroid differentiation (Martinez, P.A.; Li, R.; Ramanathan, H.N.; Bhasin, M.; Pearsall, R.S.; Kumar, R.; Suragani, R. Smad2/3-pathway ligand trap luspatercept enhances erythroid differentiation in murine beta-thalassaemia by increasing GATA-1 availability. J Cell Mol Med 202024, 6162-6177, doi:10.1111/jcmm.15243).

The exact mechanism of action of luspatercept is not fully defined and  further research is needed to clarify the consequences of TGF-beta inhibition at cellular level.

Author Response

Thanks to the reviewer for the valuable comment on the mechanism of action of Luspatercept.

The mentioned reference has already been addressed in the manuscript. However, the sentence was expanded as below;

Experimental studies using the murine analog of Luspatercept (RAP-536) demonstrated that Luspatercept-mediated inhibition of SMAD2/3 enhanced late-stage erythropoiesis by restoring nuclear levels of the transcription factor GATA-1 and up-regulation of HSP70 in erythroid precursors leading to amelioration of oxidative stress and promotion of erythroid differentiation.

The sentence below was added to the 'Current status, challenges, and future directions of luspatercept' section as below.

Further research is needed to define the exact mechanism of action of luspatercept and clarify the consequences of TGF-beta inhibition at the cellular level.

Reviewer 2 Report

Nice review, well-structured, all interesting information included

Author Response

Thanks to the reviewer for the positive feedback about the manuscript.